# Binding of Natural Antibodies Generated after COVID-19 and Vaccination with Individual Peptides Corresponding to the SARS-CoV-2 S-Protein

**DOI:** 10.3390/vaccines12040426

**Published:** 2024-04-17

**Authors:** Anna M. Timofeeva, Sergey E. Sedykh, Ekaterina A. Litvinova, Sergey A. Dolgushin, Andrey L. Matveev, Nina V. Tikunova, Georgy A. Nevinsky

**Affiliations:** 1Institute of Chemical Biology and Fundamental Medicine, Siberian Branch of the Russian Academy of Sciences, Novosibirsk 630090, Russia; sirozha@gmail.com (S.E.S.); guterus@gmail.com (A.L.M.); tikunova@niboch.nsc.ru (N.V.T.);; 2Advanced Engineering School, Novosibirsk State University, Novosibirsk 630090, Russia; 3Physical Engineering Faculty, Novosibirsk State Technical University, Novosibirsk 630073, Russia; 4Aivok LLC, Zelenograd, Moscow 124498, Russia

**Keywords:** antibody screening, vaccine, antibodies, SARS-CoV-2, S-protein, COVID-19, infectious diseases, vaccine development

## Abstract

The rapid development of vaccines is a crucial objective in modern biotechnology and molecular pharmacology. In this context, conducting research to expedite the selection of a potent immunogen is imperative. The candidate vaccine should induce the production of antibodies that can recognize the immunogenic epitopes of the target protein, resembling the ones found in recovered patients. One major challenge in vaccine development is the absence of straightforward and reliable techniques to determine the extent to which the spectrum of antibodies produced after vaccination corresponds to antibodies found after recovery. This paper describes a newly developed method to detect antibodies specific to immunogenic epitopes of the target protein in blood plasma and to compare them with antibody spectra generated post vaccination. Comparing the antibody pool generated in the human body after recovering from an infectious disease with the pool formed through vaccination can become a universal method for screening candidate vaccines. This method will enable the identification of candidate vaccines that can induce the production of antibodies similar to those generated in response to a natural infection. Implementing this approach will facilitate the rapid development of new vaccines, even when faced with a pandemic.

## 1. Introduction

The crucial aspects of vaccine creation are the rapid pace of development and the comparison and selection of the most effective immunogens. For example, when developing vaccines for COVID-19, researchers faced the challenge of identifying the most potent formulation among several candidate vaccines [1]. Studies aimed at accelerating the development of new vaccines are of significant practical value. 

B cells play a crucial role in developing a robust immune response to viral infections by producing a diverse array of antibodies that can effectively neutralize the virus [2]. However, not every viral protein fragment can trigger the generation of neutralizing antibodies. Concerning COVID-19, the immunogenic characteristics of the S-protein of the SARS-CoV-2 virus have been thoroughly researched [3]. The main immunogenic epitopes of the S-protein were identified using peptide microarrays [4]. Additionally, the Immune Epitope Database and Analysis Resource (IEDB) were created [5]. 

Identifying viral antigenic epitopes that elicit a humoral immune response is essential for developing vaccines and gaining insight into the intricacies of the immune response to viral diseases. The vaccine under development must elicit antibodies that can identify the primary epitopes of the target protein, mirroring the pattern observed in individuals who have encountered the disease. A substantial obstacle to the development of new vaccines is the lack of technologies to analyze the similarity between the immunity of vaccinated individuals and that of patients who have recovered from the disease. 

Currently, much attention is being paid to the development of epitope-based vaccines. Bioinformatics analysis methods allow one to predict potential epitopes of viral proteins. Several tools are used to predict the sequence and structure of linear epitopes using machine learning (ML) methods. EpitopeVec [6] predicts linear B cell epitopes using deep protein sequence embeddings with an accuracy greater than 80%. BepiPred-2.0 [7] has been accepted as state of the art in the community, trained on epitopes annotated from antibody–antigen protein structures; however, its accuracy is estimated to be less than 60% [6]. ABCPred [8] predicts B cell epitope regions in an antigen sequence with a prediction accuracy of approximately 65%. Thus, all methods of bioinformatics analysis only predict epitopes with some accuracy.

In addition, several experimental epitope mapping approaches can usually be divided into two types: structural and peptide [9]. Structural technologies, including X-ray crystallography [10], nuclear magnetic resonance [11], and cryo-EM [12], are the gold standards. However, it is not easy to obtain high-quality antibody–antigen crystals, which is a crucial step for successful X-ray crystallography, and it is even more complicated when the antigen is a membrane protein [13]. Peptide-based technologies include peptide microarrays [14], SPOT [15], and peptides or protein fragments displayed on phages or *E. coli* [16]. Recently, next-generation sequencing has also been introduced into this field in combination with display platforms such as phage display [17] and *E. coli* display [18]. Typically, peptides covering a single antigen or antigens are synthesized and immobilized on a planar microarray or presented on phage/*E. coli*. Peptides to which the antibody binds are then enriched and identified; typically, a large number of peptides are required, and the design of the microarray and library is complex. 

Assessment of the immunodominance of individual epitopes in the plasma of recovered patients is of significant scientific interest compared to the immune response to natural infection and vaccination [19,20]. Such an evaluation would allow the selection of the most recognizable sequences for vaccine development. This fact has motivated us to develop a system for screening the immune response to individual epitopes of viral proteins. Our screening method can be considered an additional stage for assessing the in vivo immunogenicity of epitopes predicted by bioinformatics methods. Our method is based on the S-protein epitopes of SARS-CoV-2, but it can also apply to other viral infections.

Our research aimed to create a method for screening and comparing antibodies produced against various epitopes of the SARS-CoV-2 S-protein through vaccination and natural infection. The core of the method relies on an enzyme-linked immunosorbent assay (ELISA) that utilizes oligopeptides. These oligopeptides are designed to match specific linear fragments of the S-protein sorbed in the wells of a plate. A positive ELISA signal indicates the formation of antibodies to this protein fragment. The efficiency of our method was evaluated for the preparations of monoclonal antibodies, blood plasma of mice immunized with the recombinant S-protein fragment RBD [21], Sputnik V (Gamaleya NRCEM) [22,23], and CoviVac (Chumakov FSC R&D IBP RAS) [24] vaccines, as well as of COVID-19-infected patients vaccinated with Sputnik V and CoviVac. 

Searching for epitopes that exhibit antigenicity is a crucial step in vaccine development. Comparing the pool of antibodies produced in the body during an infectious disease and due to vaccination is a promising tool to select a candidate vaccine to stimulate the production of antibodies matching the natural antigen. Thus, our approach may be alternative (or complementary) to the studies of neutralizing antibodies to identify the potential of a candidate vaccine against a virus. Additionally, our method is beneficial for the creation of epitope-based vaccines as it can assess the prominence of predicted epitopes and their appropriateness for such vaccines.

## 2. Materials and Methods

### 2.1. Oligopeptides

The sequences of nineteen biotinylated 12-mer oligopeptides corresponding to different antibody-recognized epitopes of the S-protein were selected and synthesized from those identified previously in [25]. We chose the oligopeptides located along the entire length of the S-protein, with two of them being part of the RBD (Cl, YK). The oligopeptide sequences are presented in Table 1.

The oligopeptides were synthesized by Proteogenix (Schiligeheim, France). The peptides were labeled with biotin at the C-terminus using an aminohexanoic acid linker. The quality control was performed by the manufacturer using the MS and HPLC methods.

### 2.2. Immunization of Mice and Blood Collection 

Outbred mice of the CD1 line aged three months were kept in the mouse breeding room at the premises of the Federal State Budgetary Scientific Institution “Scientific Research Institute of Neurosciences and Medicine” (Novosibirsk) under standard pathogen-free conditions. This study was conducted following the recommendations of the Declaration of Helsinki and approved by the Local Ethics Committee of the Institute of Chemical Biology and Basic Medicine (protocol of 15 August 2020).

The CD1 line mice of three groups (10 mice each) were immunized twice with 50 mkl of Sputnik V, CoviVac vaccines, or with RBD (10 mkg). The period between the first and second immunization was 21 days. The blood was collected after decapitation, 42 days after the first immunization in tubes with 4% sodium citrate (blood-to-citrate ratio was 1:9). As a control, the plasma from mice immunized with saline was used.

The plasma samples from 10 volunteers exposed to COVID-19, 10 volunteers vaccinated with Sputnik V, and 10 volunteers vaccinated with CoviVac were collected for this study. This study included individuals who received two doses of the relevant vaccine with a 21-day interval between doses. Blood collection occurred 42–63 days after the first vaccine dose or the onset of the first COVID-19 symptoms. The plasma from 10 donors who did not have COVID-19 and were not vaccinated was used as controls. All the donors signed a voluntary informed consent for the use of blood for scientific purposes. 

The tubes with blood were centrifuged at 3000× *g* for 10 min in a Centrifuge 5810 centrifuge (Eppendorf, Hamburg, Germany). The blood plasma was frozen and used to analyze the presence of antibodies.

### 2.3. Screening of Antibodies That Recognize the S-Protein Oligopeptides

An amount of 100 μL of 2 μg/mL of avidin solution in 0.1 M carbonate buffer (0.1 M NaHCO_3_, 0.1 M Na_2_CO_3_) was added to a 96-well Corning (Corning, NY, USA) high-binding microplate and incubated overnight at 4 °C. The wells of the plate were blocked with 300 μL of PBS-T with 5% non-fat dry milk (Applichem, Darmstadt, Germany) (60 min at 37 °C). The plate was washed three times with 300 μL of PBS-T, and 100 μL of 40 ng/mL of biotinylated peptide solution in PBS-T was added and incubated for 30 min. The plate was washed 3 times with 300 µL of PBS-T, and plasma diluted 1:100 in PBS-T was added to each well and incubated for 30 min with agitation. The plate was washed 3 times with 300 µL of PBS-T, and 100 µL of goat anti-human IgG antibody conjugated to horseradish peroxidase 1:10,000 was added and incubated for 30 min under stirring. The plate was washed three times with 300 μL of PBS-T. An amount of 100 µL of 3,3′,5,5′-tetramethylbenzidine was added to the wells. After 15 min, the reaction was stopped by adding 50 µL of 1 M sulfuric acid. Optical density was measured on a Multiskan FC spectrophotometer (Thermo Scientific, Waltham, MA, USA) in two-wavelength mode: main filter—450 nm, reference filter—620 nm. The results are presented as the average value of a series of three independent experiments. The measurement error did not exceed 10%.

### 2.4. Structural Visualization of Oligopeptides

The amino acid sequence of RBD was obtained from the UniProt database (https://www.uniprot.org/, accessed on 5 June 2023) [26]: P0DTC2 SPIKE_SARS2. Protein structure prediction was performed using AlphaFold2 software (ColabFold v1.5.2-patch: AlphaFold2 using MMseqs2) [27]. Molecular visualization of oligopeptides was performed using RCSB PDB Mol* 3D Viewer [28].

## 3. Results and Discussion

This work is based on the basic principles of ELISA, which have been enhanced using advances in epitope mapping. We selected epitopes of the S-protein of the SARS-CoV-2 virus based on a study in which immunodominant linear epitopes of the S-protein were identified by peptide array mapping [25]. Although we focused on these epitopes, they could be determined or predicted by any existing methods, including bioinformatics. Moreover, knowing the protein sequence of the mutant form of the virus, epitopes for it can be predicted by a bioinformatics method and further analyzed using our approach. Our approach relies on utilizing an immunoassay with biotinylated 12-mer oligopeptide sequences sorbed in plate wells [29].

The subsequent procedures are standard and include (i) the addition of test antibodies or plasma, and (ii) visualization. For colorimetric detection, antibodies are treated with secondary antibodies conjugated to a reporter enzyme such as horseradish peroxidase (or alkaline phosphatase) which catalyzes the colorimetric reaction [30,31], using 3,3′,5,5′-tetramethylbenzidine (TMB) as the substrate. Colorimetric changes can be measured using standard laboratory equipment. This study was conducted using a colorimetric detection method, which can be adapted for chemiluminescent and fluorescent detection.

A positive ELISA signal signifies that the immune system has reached the surface region of the target protein, leading to the production of antibodies. Nevertheless, advances in machine learning techniques now enable the prediction of conformational epitopes. For instance, Bepipred II [7] can accurately forecast epitopes using a random forest algorithm [32]. Our method also allows such epitopes to be analyzed. 

The first step was to analyze the binding of five monoclonal antibodies (mAbs) (RS1, RS2, RS17, RS18, and RS21) against the S-protein of SARS-CoV-2 virus to nineteen oligopeptide sequences using the developed platform (Figure 1A). Three of the five mAbs were found to bind to only one oligopeptide sequence: RS1 to YK oligopeptide, RS2 to AR, RS18 to DV, and two mAbs did not bind to any of the sequences tested. Remarkably, we observed negligible optical density in some plate wells, which can be explained by nonspecific antibody binding. 

As expected, the monoclonal antibodies did not bind to more than one oligopeptide sequence. The epitope recognized by mAb RS17 is located at the N-terminal site of the RBD (348-SVYAVNAVNRKRIS-358) [33]. This epitope is not within the range of oligopeptide sequences tested, so it is not surprising that no positive ELISA signal was observed for this monoclonal antibody. The characterization of putative epitopes for the remaining mAbs has been exclusively achieved through our system, as opposed to other methods. 

The second step was the analysis of polyclonal antibodies produced through the immunization of mice or vaccination of humans; then, we screened the blood plasma of mice immunized with RBD, Sputnik V, and CoviVac vaccines. The blood plasma of mice immunized with saline was used as a control. Figure 1B illustrates the findings. The oligopeptides CI and YK are known to be part of the RBD, and antibodies recognizing only the YK oligopeptide were observed in mice immunized with RBD. 

AlphaFold2 software can predict protein structure with atomic precision; this method can compete even with crystallographic methods [34,35]. These results inspired us to use this tool to visualize epitopes in protein structure. The RBD structure was visualized using AlphaFold2 software (ColabFold v1.5.2-patch: AlphaFold2 using MMseqs2) [27] and RCSB PDB Mol* 3D Viewer [28]. The visualization revealed the YK amino acid sequence to be situated on the surface of the RBD, suggesting that it could be a recognizable site on the RBD surface. The RBD is folded in a manner that prevents the immune system from recognizing the amino acid sequence associated with CI. Therefore, it is natural that antibodies recognizing the CI oligopeptide are not detected in mouse plasma (Figure 1C). A slightly positive ELISA signal to different oligopeptide sequences indicates a tendency for nonspecific cross-reactivity with other amino acid sequences.

Mice immunized with Sputnik V and CoviVac vaccines were found to have antibodies recognizing oligopeptide YK, a component of RBD. Mice immunized with Sputnik V were detected to have antibodies recognizing oligopeptides PV, QF, KL, and QG. Mice immunized with CoviVac were identified to have antibodies recognizing PV, QF, KL, and AR. It should be highlighted that the antibody profiles observed in mice immunized with the two vaccines under investigation were similar. 

The third stage was to screen plasma from people infected with COVID-19 and vaccinated with Sputnik V and CoviVac. The blood plasma of donors without COVID-19 and unvaccinated was used as a control. The total blood plasma preparation of 10 donors was used for analysis. The optical density values in the control sample were comparable to the background values. The results are presented in Figure 1D. The plasma of COVID-19-transfected donors was found to contain antibodies recognizing all of the 19 oligopeptides. These results are not unexpected, since the oligopeptides corresponding to the most recognized epitopes of the S-protein were initially selected. The oligopeptides most efficiently recognized by antibodies of immunized humans include QF, YK, FI, PQ, PV, GE, and EA.

Thus, the profiles of antibodies formed during immunization with Sputnik V and CoviVac were found to be similar. The oligopeptides YK, PV, PQ, QF, and KL, predominantly recognized by antibodies during immunization with both vaccines, also proved to be similar. Plasma antibodies from vaccinated patients, as compared to those from COVID-19 survivors, showed weak specificity for oligopeptides GE, EA, and FI. 

Both COVID-19 patients and vaccinated donors’ antibodies had little or no specificity for oligopeptides DK and FA. This finding is consistent with the result of oligopeptide visualization on the S-protein model, indicating that these oligopeptide sequences are located deep within the S-protein and are unlikely to be recognized by the immune system (see Figure 2). The visualization of oligopeptides demonstrates that the amino acid sequences corresponding to DV, KL, and TQ oligopeptides are located on the surface of the S-protein and are available for recognition by the immune system. Interestingly, antibodies recognizing DV, KL, and TQ oligopeptides were found to be formed during vaccination. However, such antibodies were barely detected after COVID-19.

Our strategy opens the prospect of rapid mapping of the antigenic properties of monoclonal and polyclonal antibodies. Our approach demonstrated its effectiveness in analyzing monoclonal antibodies with only one detectable binding site. The screening process allowed us to establish that immunization using both Sputnik V and CoviVac vaccines resulted in comparable antibody profiles. Immunization with these vaccines produces similar antibody profiles to those resulting from transferred COVID-19. It should be noted that a robust immune response to some epitopes does not necessarily result in the effective neutralizing activity of these antibodies against the virus. In summary, the method developed allows the analysis of monoclonal antibodies, the determination of antibodies in blood plasma specific to certain epitopes of the target protein, and the comparison with antibodies formed during vaccination. 

## 4. Conclusions

Identifying viral B-cell epitopes is essential in selecting peptides for inclusion in subunit vaccines, developing virus-specific serological tests, and understanding antibody–virus interactions at the molecular level. B cell epitopes can be predicted in various ways, including bioinformatics; however, such methods do not guarantee high accuracy of the result. Other methods (deep mutation scanning, peptide/protein microarrays, bacteriophage peptide/protein display) are expensive and time-consuming. Therefore, we developed a simple approach to assess the immunogenicity of selected epitopes. This method can be used as an additional screening stage after bioinformatics analysis.

The approach described in this study can determine the presence in the plasma of antibodies specific to immunogenic epitopes of the target protein and can compare them with antibodies formed during vaccination. Comparing the pool of antibodies formed in the body after an infectious disease and due to vaccination is a convenient tool for screening the potential of vaccines and selecting those that stimulate the production of antibodies corresponding to the natural infectious agent. Such screening, aimed at identifying the potential of a candidate vaccine, will contribute to accelerating the development timeframe for new vaccines, including in pandemic settings. Of great importance for the development of currently promising epitope-based vaccines is that our method allows one to identify the most promising epitopes of viral proteins from among those predicted by bioinformatics analysis methods.

## Figures and Tables

**Figure 1 vaccines-12-00426-f001:**
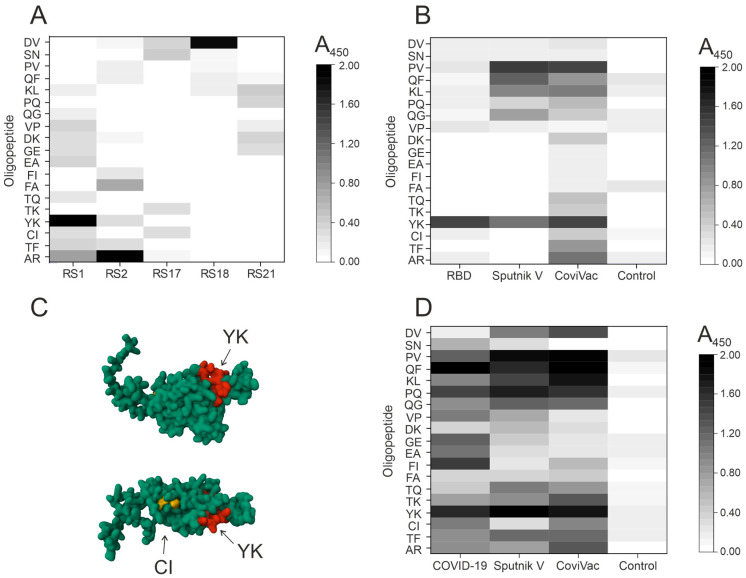
Screening of binding of five mAbs (RS1, RS2, RS17, RS18, and RS21) to 12-mer oligopeptides that are epitopes of SARS-CoV-2 S protein (**A**). Screening of blood plasma of mice immunized with RBD and vaccines for the presence of antibodies recognizing various epitopes of the SARS-CoV-2 S protein (**B**). Visualization of OPs corresponding to the amino acid residues of the RBD was performed in AlphaFold2 software (ColabFold v1.5.2-patch: AlphaFold2 using MMseqs2) [27] and RCSB PDB Mol* 3D Viewer [28]: two views are shown. The amino acid sequence of the YK oligopeptide is indicated in red, and CI is indicated in yellow (**C**). Screening of blood plasma of patients who recovered from COVID-19 and were immunized with vaccines for the presence of antibodies recognizing various epitopes of the SARS-CoV-2 S protein (**D**).

**Figure 2 vaccines-12-00426-f002:**
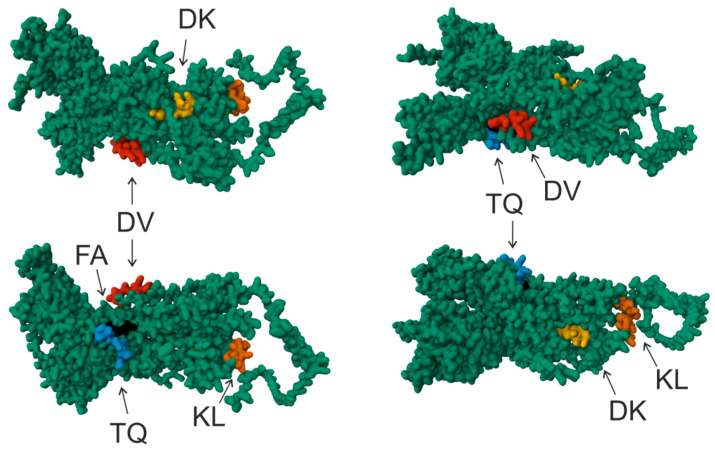
Visualization of OPs corresponding to the amino acid residues of the RBD was performed in AlphaFold2 software (ColabFold v1.5.2-patch: AlphaFold2 using MMseqs2) [27] and RCSB PDB Mol* 3D Viewer [28]. Four projections are given. The amino acid sequence of the DV oligopeptide is shown in red, KL in orange, TQ in blue, DK in yellow, and FA in black.

**Table 1 vaccines-12-00426-t001:** Oligopeptides used for immunization of laboratory animals.

Oligopeptide Designation	Amino Acid Residues in the SARS-CoV-2 S-Protein	Sequence of the Oligopeptide Amino Acid Residues
AR	67–78	AIHVSGTNGTKR
TF	307–318	TVEKGIYQTSNF
CI	391–402	CFTNVYADSFVI
YK	451–462	YLYRLFRKSNLK
TK	547–558	TGTGVLTESNKK
TQ	553–564	TESNKKFLPFQQ
FA	559–570	FLPFQQFGRDIA
FI	655–666	FVNNSYECDIPI
EA	661–672	ECDIPIGAGICA
GE	769–780	GIAVEQDKNTQE
DK	775–786	DKNTQEVFAQVK
VP	781–792	VFAQVKQIYKTP
QG	787–798	QIYKTPPIKDFG
PQ	793–804	PIKDFGGFNFSQ
KL	811–822	KPSKRSFIEDLL
QF	895–906	QIPFAMQMAYRF
PV	1057–1068	PHGVVFLHVTYV
SN	1147–1158	SFKEELDKYFKN
DV	1153–1164	DKYFKNHTSPDV

## Data Availability

Empirical data that do not relate to the personal data of donors can be provided by request to Anna Timofeeva.

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
