# Peer review of "Binding of Natural Antibodies Generated after COVID-19 and Vaccination with Individual Peptides Corresponding to the SARS-CoV-2 S-Protein"

_vaccines, 2024, doi:10.3390/vaccines12040426_

Round 1

Reviewer 1 Report

Comments and Suggestions for Authors

I would suggest strengthening the introduction by highlighting the specific challenges in vaccine development and how the current methods (including yours and others') address them. 

Author Response

We have expanded the Introduction and focused on methods. Please, see lines 43–48 and 58–85.

Reviewer 2 Report

Comments and Suggestions for Authors

  It is important to find immunodominant epitopes that elicit effective neutralizing antibodies in the vaccine development. In the current study, the authors analyzed polyclonal antibodies elicited through COVID-19 vaccination and compared them with the antibody pool produced in the human body during natural infection. Interestingly, it seems the antibody pattern of recovered patients is not identical, but similar to that of individuals immunized with Sputnik V or CovVac. Because the bioinformatics method hardly shows the immunodominance of predicted peptides, this approach may be a convenient tool for screening the potential of candidate vaccines. The idea is very interesting and the data contain useful information. Although the manuscript is well-written, I have raised two points which need to be clarified. There are given below.

Specific comments:

1)     The authors should discuss the influence of mutations in SARS-CoV-2 on this approach.

2)     In Fig. 1-D, there are several oligopeptides that were strongly recognized by the antibody pool derived from recovered patients. However, I think strongly reacted epitopes in this analysis do not necessarily induce effective neutralizing antibodies. The authors should discuss it.

Author Response

It is important to find immunodominant epitopes that elicit effective neutralizing antibodies in the vaccine development. In the current study, the authors analyzed polyclonal antibodies elicited through COVID-19 vaccination and compared them with the antibody pool produced in the human body during natural infection. Interestingly, it seems the antibody pattern of recovered patients is not identical, but similar to that of individuals immunized with Sputnik V or CovVac. Because the bioinformatics method hardly shows the immunodominance of predicted peptides, this approach may be a convenient tool for screening the potential of candidate vaccines. The idea is very interesting and the data contain useful information. Although the manuscript is well-written, I have raised two points which need to be clarified. There are given below.

Specific comments:

1) The authors should discuss the influence of mutations in SARS-CoV-2 on this approach.

We've added this discussion, please, see lines 116–119

2) In Fig. 1-D, there are several oligopeptides that were strongly recognized by the antibody pool derived from recovered patients. However, I think strongly reacted epitopes in this analysis do not necessarily induce effective neutralizing antibodies. The authors should discuss it.

We agree with the reviewer that a strong immune response against specific epitopes does not necessarily result in the effective neutralizing activity of these antibodies against the virus. We noted this in the work; please, see lines 228–230. However, in this work, we did not study the neutralizing activity of antibodies.

Reviewer 3 Report

Comments and Suggestions for Authors

1. Despite the authors' statement (Introduction) that "... these studies do not evaluate the immunodominance of predicted epitopes in the plasma of re-infected patients" there are published studies, which are initially based on the determination of immunodominant epitopes of viral proteins either by using infected patient plasma or immunized animal sera and T cell preparations (e.g. Belyavtsev et al., Bioorg. Chem., 2021, 47(3), 713-718; Volkova et al., Bioorg. Khim., 2007,  33(2), 229-234, Chibiskova et al.,  Bull Exp Biol Med. 2007,143(6):720-722 and others). At least some of such papers should be cited. 

2. The procedure of animal presentation is not described in details. Doses of vaccines are not shown - are they the same as for humans or not? What was the course of the vaccination? Time intervals between the vaccination (or COVID 19 infection) and blood collection are not defined. 

3. No details of biotinylated peptide structures are defined. What linkers are  employed between the S-protein fragment sequence and biotin residue? Is biotin attached to N- or C-termini of the peptides? Biotin-binding proteins have rather deep pockets for biotin binding, hence biotin is usually attached to peptides via rather long linkers, otherwise 3-4 amino acid residues close to biotin will not participate in antibody binding because of their immersion into the protein pocket. The absence of such a linker will change the results of the epitope determination and their discussion.       

Author Response

  1. Despite the authors' statement (Introduction) that "... these studies do not evaluate the immunodominance of predicted epitopes in the plasma of re-infected patients" there are published studies, which are initially based on the determination of immunodominant epitopes of viral proteins either by using infected patient plasma or immunized animal sera and T cell preparations (e.g. Belyavtsev et al., Bioorg. Chem., 2021, 47(3), 713-718; Volkova et al., Bioorg. Khim., 2007,  33(2), 229-234, Chibiskova et al.,  Bull Exp Biol Med. 2007,143(6):720-722 and others). At least some of such papers should be cited. 

We added these references: please, see Ref 23 and Ref 24

  1. The procedure of animal presentation is not described in details. Doses of vaccines are not shown - are they the same as for humans or not? What was the course of the vaccination? Time intervals between the vaccination (or COVID 19 infection) and blood collection are not defined.

We've added these details in the manuscript. Please, see the lines 253–257.

  1. No details of biotinylated peptide structures are defined. What linkers are  employed between the S-protein fragment sequence and biotin residue? Is biotin attached to N- or C-termini of the peptides? Biotin-binding proteins have rather deep pockets for biotin binding, hence biotin is usually attached to peptides via rather long linkers, otherwise 3-4 amino acid residues close to biotin will not participate in antibody binding because of their immersion into the protein pocket. The absence of such a linker will change the results of the epitope determination and their discussion. 

The peptides were labeled with biotin at the C-terminus using an aminohexanoic acid (Ahx) linker. We have added the information. See lines 243–244.

Round 2

Reviewer 3 Report

Comments and Suggestions for Authors

A new version of the manuscript contains some information of common knowledge for specialists in vaccines in the Introduction section: lines 43-48; 76-85; 113-114; 116-119; 121-126; 139-141; 165-166;175-176. This information can be excluded without influencing the value of the manuscript and in order to reduce the too lengthy section.  The statement in lines 291-298 somewhat repeats the information in the Introduction and shoud be excluded or changed.  

Author Response

Dear Reviewer!

We have tried to eliminate all the shortcomings that you pointed out in your comments.

1) We removed lines 43-48; 78-85; 121-126; 139-141; 165-166; 175-176, since they contained trivial information

2) We left the lines 116-119, since the Reviewer 2 asked to add this information to the article.

We believe the manuscript's modified form has become more interesting for potential readers.

Thank you on the behalf of co-authors of the manuscript!